# On Flat versus Hierarchical Classification in Large-Scale Taxonomies

**Rohit Babbar, Ioannis Partalas, Eric Gaussier, Massih-Reza Amini**
Université Joseph Fourier, Laboratoire Informatique de Grenoble
BP 53 - F-38041 Grenoble Cedex 9
`firstname.lastname@imag.fr`

## Abstract

We study in this paper flat and hierarchical classification strategies in the context of large-scale taxonomies. To this end, we first propose a multiclass, hierarchical data dependent bound on the generalization error of classifiers deployed in large-scale taxonomies. This bound provides an explanation to several empirical results reported in the literature, related to the performance of flat and hierarchical classifiers. We then introduce another type of bound targeting the approximation error of a family of classifiers, and derive from it features used in a meta-classifier to decide which nodes to prune (or flatten) in a large-scale taxonomy. We finally illustrate the theoretical developments through several experiments conducted on two widely used taxonomies.

## 1 Introduction

Large-scale classification of textual and visual data into a large number of target classes has been the focus of several studies, from researchers and developers in industry and academia alike. The target classes in such large-scale scenarios typically have an inherent hierarchical structure, usually in the form of a rooted tree, as in Directory Mozilla[1], or a directed acyclic graph, with a parent-child relationship. Various classification techniques have been proposed for deploying classifiers in such large-scale taxonomies, from flat (sometimes referred to as *big bang*) approaches to fully hierarchical one adopting a complete top-down strategy. Several attempts have also been made in order to develop new classification techniques that integrate, at least partly, the hierarchy into the objective function being optimized (as [3, 5, 10, 11] among others). These techniques are however costly in practice and most studies either rely on a flat classifier, or a hierarchical one either deployed on the original hierarchy or a simplified version of it obtained by pruning some nodes (as [15, 18])[2].

Hierarchical models for large scale classification however suffer from the fact that they have to make many decisions prior to reach a final category. This intermediate decision making leads to the error propagation phenomenon causing a decrease in accuracy. On the other hand, flat classifiers rely on a single decision including all the final categories, a single decision that is however difficult to make as it involves many categories, potentially unbalanced. It is thus very difficult to assess which strategy is best and there is no consensus, at the time being, on to which approach, flat or hierarchical, should be preferred on a particular category system.

In this paper, we address this problem and introduce new bounds on the generalization errors of classifiers deployed in large-scale taxonomies. These bounds make explicit the trade-off that both flat and hierarchical classifiers encounter in large-scale taxonomies and provide an explanation to

several empirical findings reported in previous studies. To our knowledge, this is the first time that such bounds are introduced and that an explanation of the behavior of flat and hierarchical classifiers is based on theoretical grounds. We also propose a well-founded way to select nodes that should be pruned so as to derive a taxonomy better suited to the classification problem. Contrary to [4] that reweighs the edges in a taxonomy through a cost sensitive loss function to achieve this goal, we use here a simple pruning strategy that modifies the taxonomy in an explicit way.

The remainder of the paper is organized as follows: Section 2 introduces the notations used and presents the generalization error bounds for classification in large-scale taxonomies. It also presents the meta-classifier we designed to select those nodes that should be pruned in the original taxonomy. Section 3 illustrates these developments via experiments conducted on several taxonomies extracted from DMOZ and the International Patent Classification. The experimental results are in line with results reported in previous studies, as well as with our theoretical developments. Finally, Section 4 concludes this study.

## 2 Generalization Error Analyses

Let $\mathcal{X} \subseteq \mathbb{R}^d$ be the input space and let $V$ be a finite set of class labels. We further assume that examples are pairs $(\mathbf{x}, v)$ drawn according to a fixed but unknown distribution $\mathcal{D}$ over $\mathcal{X} \times V$. In the case of hierarchical classification, the hierarchy of classes $\mathcal{H} = (V, E)$ is defined in the form of a rooted tree, with a root $\perp$ and a parent relationship $\pi : V \setminus \{\perp\} \to V$ where $\pi(v)$ is the parent of node $v \in V \setminus \{\perp\}$, and $E$ denotes the set of edges with parent to child orientation. For each node $v \in V \setminus \{\perp\}$, we further define the set of its sisters $\mathfrak{S}(v) = \{v' \in V \setminus \{\perp\}; v \neq v' \wedge \pi(v) = \pi(v')\}$ and its daughters $\mathfrak{D}(v) = \{v' \in V \setminus \{\perp\}; \pi(v') = v\}$. The nodes at the intermediary levels of the hierarchy define general class labels while the specialized nodes at the leaf level, denoted by $\mathcal{Y} = \{y \in V : \nexists v \in V, (y, v) \in E\} \subset V$, constitute the set of target classes. Finally for each class $y$ in $\mathcal{Y}$ we define the set of its ancestors $\mathfrak{P}(y)$ defined as

$$\mathfrak{P}(y) = \{v_1^y, \ldots, v_{k_y}^y; v_1^y = \pi(y) \wedge \forall l \in \{1, \ldots, k_y - 1\}, v_{l+1}^y = \pi(v_l^y) \wedge \pi(v_{k_y}^y) = \perp\}$$

For classifying an example $\mathbf{x}$, we consider a top-down classifier making decisions at each level of the hierarchy, this process sometimes referred to as the *Pachinko* machine selects the best class at each level of the hierarchy and iteratively proceeds down the hierarchy. In the case of flat classification, the hierarchy $\mathcal{H}$ is ignored, $\mathcal{Y} = V$, and the problem reduces to the classical supervised multiclass classification problem.

### 2.1 A hierarchical Rademacher data-dependent bound

Our main result is the following theorem which provides a data-dependent bound on the generalization error of a top-down multiclass hierarchical classifier. We consider here kernel-based hypotheses, with $K : \mathcal{X} \times \mathcal{X} \to \mathbb{R}$ a PDS kernel and $\Phi : \mathcal{X} \to \mathbb{H}$ its associated feature mapping function, defined as :

$$\mathcal{F}_B = \{f : (\mathbf{x}, v) \in \mathcal{X} \times V \mapsto \langle \Phi(\mathbf{x}), \mathbf{w}_v \rangle \mid \mathbf{W} = (w_1 \ldots, w_{|V|}), ||\mathbf{W}||_{\mathbb{H}} \leq B\}$$

where $\mathbf{W} = (w_1 \ldots, w_{|V|})$ is the matrix formed by the $|V|$ weight vectors defining the kernel-based hypotheses, $\langle ., . \rangle$ denotes the dot product, and $||\mathbf{W}||_{\mathbb{H}} = \left(\sum_{v \in V} ||\mathbf{w}_v||^2\right)^{1/2}$ is the $L_{\mathbb{H}}^2$ group norm of $\mathbf{W}$. We further define the following associated function class:

$$\mathcal{G}_{\mathcal{F}_B} = \{g_f : (\mathbf{x}, y) \in \mathcal{X} \times \mathcal{Y} \mapsto \min_{v \in \mathfrak{P}(y)} (f(\mathbf{x}, v) - \max_{v' \in \mathfrak{S}(v)} f(\mathbf{x}, v')) \mid f \in \mathcal{F}_B\}$$

For a given hypothesis $f \in \mathcal{F}_B$, the sign of its associated function $g_f \in \mathcal{G}_{\mathcal{F}_B}$ directly defines a hierarchical classification rule for $f$ as the top-down classification scheme outlined before simply amounts to: *assign* $\mathbf{x}$ *to* $y$ *iff* $g_f(\mathbf{x}, y) > 0$. The learning problem we address is then to find a hypothesis $f$ from $\mathcal{F}_B$ such that the generalization error of $g_f \in \mathcal{G}_{\mathcal{F}_B}$, $\mathcal{E}(g_f) = \mathbb{E}_{(\mathbf{x},y) \sim \mathcal{D}}\left[\mathbf{1}_{g_f(\mathbf{x},y) \leq 0}\right]$, is minimal ($\mathbf{1}_{g_f(\mathbf{x},y) \leq 0}$ is the 0/1 loss, equal to 1 if $g_f(\mathbf{x}, y) \leq 0$ and 0 otherwise).

The following theorem sheds light on the trade-off between flat versus hierarchical classification. The notion of function class capacity used here is the *empirical Rademacher complexity* [1]. The proof of the theorem is given in the supplementary material.

**Theorem 1** *Let $\mathcal{S} = ((\mathbf{x}^{(i)}, y^{(i)}))_{i=1}^m$ be a dataset of $m$ examples drawn i.i.d. according to a probability distribution $\mathcal{D}$ over $\mathcal{X} \times \mathcal{Y}$, and let $\mathcal{A}$ be a Lipschitz function with constant $L$ dominating the $0/1$ loss; further let $K : \mathcal{X} \times \mathcal{X} \to \mathbb{R}$ be a PDS kernel and let $\Phi : \mathcal{X} \to \mathbb{H}$ be the associated feature mapping function. Assume that there exists $R > 0$ such that $K(\mathbf{x}, \mathbf{x}) \leq R^2$ for all $\mathbf{x} \in \mathcal{X}$. Then, for all $1 > \delta > 0$, with probability at least $(1 - \delta)$ the following hierarchical multiclass classification generalization bound holds for all $g_f \in \mathcal{G}_{\mathcal{F}_B}$ :*

$$\mathcal{E}(g_f) \leq \frac{1}{m} \sum_{i=1}^m \mathcal{A}(g_f(\mathbf{x}^{(i)}, y^{(i)})) + \frac{8BRL}{\sqrt{m}} \sum_{v \in V \setminus \mathcal{Y}} |\mathfrak{D}(v)|(|\mathfrak{D}(v)| - 1) + 3\sqrt{\frac{\ln(2/\delta)}{2m}} \quad (1)$$

*where $|\mathfrak{D}(v)|$ denotes the number of daughters of node $v$.*

For flat multiclass classification, we recover the bounds of [12] by considering a hierarchy containing a root node with as many daughters as there are categories. Note that the definition of functions in $\mathcal{G}_{\mathcal{F}_B}$ subsumes the definition of the margin function used for the flat multiclass classification problems in [12], and that the factor $8L$ in the complexity term of the bound, instead of $4$ in [12], is due to the fact that we are using an $L$-Lipschitz loss function dominating the $0/1$ loss in the empirical Rademacher complexity.

**Flat vs hierarchical classification on large-scale taxonomies.** The generalization error is controlled in inequality (1) by a trade-off between the empirical error and the Rademacher complexity of the class of classifiers. The Rademacher complexity term favors hierarchical classifiers over flat ones, as any split of a set of category of size $n$ in $k$ parts $n_1, \cdots, n_k$ ($\sum_{i=1}^k n_i = n$) is such that $\sum_{i=1}^k n_i^2 \leq n^2$. On the other hand, the empirical error term is likely to favor flat classifiers vs hierarchical ones, as the latter rely on a series of decisions (as many as the length of the path from the root to the chosen category in $\mathcal{Y}$) and are thus more likely to make mistakes. This fact is often referred to as the *propagation error* problem in hierarchical classification.

On the contrary, flat classifiers rely on a single decision and are not prone to this problem (even though the decision to be made is harder). When the classification problem in $\mathcal{Y}$ is highly unbalanced, then the decision that a flat classifier has to make is difficult; hierarchical classifiers still have to make several decisions, but the imbalance problem is less severe on each of them. So, in this case, even though the empirical error of hierarchical classifiers may be higher than the one of flat ones, the difference can be counterbalanced by the Rademacher complexity term, and the bound in Theorem 1 suggests that hierarchical classifiers should be preferred over flat ones.

On the other hand, when the data is well balanced, the Rademacher complexity term may not be sufficient to overcome the difference in empirical errors due to the propagation error in hierarchical classifiers; in this case, Theorem 1 suggests that flat classifiers should be preferred to hierarchical ones. These results have been empirically observed in different studies on classification in large-scale taxonomies and are further discussed in Section 3.

Similarly, one way to improve the accuracy of classifiers deployed in large-scale taxonomies is to modify the taxonomy by pruning (sets of) nodes [18]. By doing so, one is flattening part of the taxonomy and is once again trading-off the two terms in inequality (1): pruning nodes leads to reduce the number of decisions made by the hierarchical classifier while maintaining a reasonable Rademacher complexity. Even though it can explain several empirical results obtained so far, the bound displayed in Theorem 1 does not provide a practical way to decide on whether to prune a node or not, as it would involve the training of many classifiers which is impractical with large-scale taxonomies. We thus turn towards another bound in the next section that will help us design a direct and simple strategy to prune nodes in a taxonomy.

## 2.2 Asymptotic approximation error bounds

We now propose an asymptotic approximation error bound for a multiclass logistic regression (MLR) classifier. We first consider the flat, multiclass case ($V = \mathcal{Y}$), and then show how the bounds can be combined in a typical top-down cascade, leading to the identification of important features that control the variation of these bounds.

Considering a pivot class $y^\star \in \mathcal{Y}$, a MLR classifier, with parameters $\boldsymbol{\beta} = \{\beta_0^y, \beta_j^y; y \in \mathcal{Y} \setminus \{y^\star\}, j \in \{1, \ldots, d\}\}$, models the class posterior probabilities via a linear function in $\mathbf{x} = (x_j)_{j=1}^d$ (see for example [13] p. 96) :

$$
\begin{aligned}
P(y|\mathbf{x}; \boldsymbol{\beta})_{y \neq y^\star} &= \frac{\exp(\beta_0^y + \sum_{j=1}^d \beta_j^y x_j)}{1 + \sum_{y' \in \mathcal{Y}, y' \neq y^\star} \exp(\beta_0^{y'} + \sum_{j=1}^d \beta_j^{y'} x_j)} \\
P(y^\star|\mathbf{x}; \boldsymbol{\beta}) &= \frac{1}{1 + \sum_{y' \in \mathcal{Y}, y' \neq y^\star} \exp(\beta_0^{y'} + \sum_{j=1}^d \beta_j^{y'} x_j)}
\end{aligned}
$$

The parameters $\boldsymbol{\beta}$ are usually fit by maximum likelihood over a training set $\mathcal{S}$ of size $m$ (denoted by $\widehat{\boldsymbol{\beta}}_m$ in the following) and the decision rule for this classifier consists in choosing the class with the highest class posterior probability :

$$
h_m(\mathbf{x}) = \underset{y \in \mathcal{Y}}{\operatorname{argmax}} \, P(y|\mathbf{x}, \widehat{\boldsymbol{\beta}}_m) \tag{2}
$$

The following lemma states to which extent the posterior probabilities with maximum likelihood estimates $\widehat{\boldsymbol{\beta}}_m$ may deviate from their asymptotic values obtained with maximum likelihood estimates when the training size $m$ tends to infinity (denoted by $\widehat{\boldsymbol{\beta}}_\infty$).

**Lemma 1** *Let $\mathcal{S}$ be a training set of size $m$ and let $\widehat{\boldsymbol{\beta}}_m$ be the maximum likelihood estimates of the MLR classifier over $\mathcal{S}$. Further, let $\widehat{\boldsymbol{\beta}}_\infty$ be the maximum likelihood estimates of parameters of MLR when $m$ tends to infinity. For all examples $\mathbf{x}$, let $R > 0$ be the bound such that $\forall y \in \mathcal{Y} \setminus \{y^\star\}, \exp(\beta_0^y + \sum_{j=1}^d \beta_j^y x_j) < \sqrt{R}$; then for all $1 > \delta > 0$, with probability at least $(1 - \delta)$ we have:*

$$
\forall y \in \mathcal{Y}, \left| P(y|\mathbf{x}, \widehat{\boldsymbol{\beta}}_m) - P(y|\mathbf{x}, \widehat{\boldsymbol{\beta}}_\infty) \right| < d\sqrt{\frac{R|\mathcal{Y}|\sigma_0}{\delta m}}
$$

*where $\sigma_0 = \max_{j,y} \sigma_j^y$ and $(\sigma_j^y)_{y,j}$ represent the components of the inverse (diagonal) Fisher information matrix at $\widehat{\boldsymbol{\beta}}_\infty$.*

**Proof (sketch)** By denoting the sets of parameters $\widehat{\boldsymbol{\beta}}_m = \{\hat{\beta}_j^y; j \in \{0, \ldots, d\}, y \in \mathcal{Y} \setminus \{y^\star\}\}$, and $\widehat{\boldsymbol{\beta}}_\infty = \{\beta_j^y; j \in \{0, \ldots, d\}, y \in \mathcal{Y} \setminus \{y^\star\}\}$, and using the independence assumption and the asymptotic normality of maximum likelihood estimates (see for example [17], p. 421), we have, for $0 \leq j \leq d$ and $\forall y \in \mathcal{Y} \setminus \{y^\star\}$: $\sqrt{m}(\hat{\beta}_j^y - \beta_j^y) \sim N(0, \sigma_j^y)$ where the $(\sigma_j^y)_{y,i}$ represent the components of the inverse (diagonal) Fisher information matrix at $\widehat{\boldsymbol{\beta}}_\infty$. Let $\sigma_0 = \max_{j,y} \sigma_j^y$. Then using Chebyshev's inequality, for $0 \leq j \leq d$ and $\forall y \in \mathcal{Y} \setminus \{y^\star\}$ we have with probability at least $1 - \sigma_0/\epsilon^2$, $|\hat{\beta}_j^y - \beta_j^y| < \frac{\epsilon}{\sqrt{m}}$. Further $\forall \mathbf{x}$ and $\forall y \in \mathcal{Y} \setminus \{y^\star\}, \exp(\beta_0^y + \sum_{j=1}^d \beta_j^y x_j) < \sqrt{R}$; using a Taylor development of the functions $\exp(x+\epsilon)$ and $(1+x+\epsilon x)^{-1}$ and the union bound, one obtains that, $\forall \epsilon > 0$ and $y \in \mathcal{Y}$ with probability at least $1 - \frac{|\mathcal{Y}|\sigma_0}{\epsilon^2}$: $\left| P(y|\mathbf{x}, \widehat{\boldsymbol{\beta}}_m) - P(y|\mathbf{x}, \widehat{\boldsymbol{\beta}}_\infty) \right| < d\sqrt{\frac{R}{m}}\epsilon$. Setting $\frac{|\mathcal{Y}|\sigma_0}{\epsilon^2}$ to $\delta$, and solving for $\epsilon$ gives the result. $\square$

Lemma 1 suggests that the predicted and asymptotic posterior probabilities are close to each other, as the quantities they are based on are close to each other. Thus, provided that the asymptotic posterior probabilities between the best two classes, for any given $\mathbf{x}$, are not too close to each other, the generalization error of the MLR classifier and the one of its asymptotic version should be similar. Theorem 2 below states such a relationship, using the following function that measures the confusion between the best two classes for the asymptotic MLR classifier defined as :

$$
h_\infty(\mathbf{x}) = \underset{y \in \mathcal{Y}}{\operatorname{argmax}} \, P(y|\mathbf{x}, \widehat{\boldsymbol{\beta}}_\infty) \tag{3}
$$

For any given $\mathbf{x} \in \mathcal{X}$, the confusion between the best two classes is defined as follows.

**Definition 1** *Let $f_\infty^1(\mathbf{x}) = \max_{y \in \mathcal{Y}} P(y|\mathbf{x}, \widehat{\boldsymbol{\beta}}_\infty)$ be the best class posterior probability for $\mathbf{x}$ by the asymptotic* MLR *classifier, and let $f_\infty^2(\mathbf{x}) = \max_{y \in \mathcal{Y} \setminus h_\infty(\mathbf{x})} P(y|\mathbf{x}, \widehat{\boldsymbol{\beta}}_\infty)$ be the second best class posterior probability for $\mathbf{x}$. We define the confusion of the asymptotic* MLR *classifier for a category set $\mathcal{Y}$ as:*

$$G_{\mathcal{Y}}(\tau) = P_{(\mathbf{x},y) \sim \mathcal{D}}(|f_\infty^1(\mathbf{x}) - f_\infty^2(\mathbf{x})| < 2\tau)$$

*for a given $\tau > 0$.*

The following theorem states a relationship between the generalization error of a trained MLR classifier and its asymptotic version.

**Theorem 2** *For a multi-class classification problem in $d$ dimensional feature space with a training set of size $m$, $\{\boldsymbol{x}^{(i)}, y^{(i)}\}_{i=1}^{m}$, $\boldsymbol{x}^{(i)} \in \mathcal{X}$, $y^{(i)} \in \mathcal{Y}$, sampled i.i.d. from a probability distribution $\mathcal{D}$, let $h_m$ and $h_\infty$ denote the multiclass logistic regression classifiers learned from a training set of finite size $m$ and its asymptotic version respectively, and let $\mathcal{E}(h_m)$ and $\mathcal{E}(h_\infty)$ be their generalization errors. Then, for all $1 > \delta > 0$, with probability at least $(1 - \delta)$ we have:*

$$\mathcal{E}(h_m) \le \mathcal{E}(h_\infty) + G_{\mathcal{Y}}\left(d\sqrt{\frac{R|\mathcal{Y}|\sigma_0}{\delta m}}\right) \qquad (4)$$

*where $\sqrt{R}$ is a bound on the function $\exp(\beta_0^y + \sum_{j=1}^{d} \beta_j^y x_j)$, $\forall \boldsymbol{x} \in \mathcal{X}$ and $\forall y \in \mathcal{Y}$, and $\sigma_0$ is a constant.*

**Proof (sketch)** The difference $\mathcal{E}(h_m) - \mathcal{E}(h_\infty)$ is bounded by the probability that the asymptotic MLR classifier $h_\infty$ correctly classifies an example $(\mathbf{x}, y) \in \mathcal{X} \times \mathcal{Y}$ randomly chosen from $\mathcal{D}$, while $h_m$ misclassifies it. Using Lemma 1, for all $\delta \in (0, 1), \forall \mathbf{x} \in \mathcal{X}, \forall y \in \mathcal{Y}$, with probability at least $1 - \delta$, we have:

$$\left| P(y|\mathbf{x}, \widehat{\boldsymbol{\beta}}_m) - P(y|\mathbf{x}, \widehat{\boldsymbol{\beta}}_\infty) \right| < d\sqrt{\frac{R|\mathcal{Y}|\sigma_0}{\delta m}}$$

Thus, the decision made by the trained MLR and its asymptotic version on an example $(\mathbf{x}, y)$ differs only if the distance between the two predicted classes of the asymptotic classifier is less than two times the distance between the posterior probabilities obtained with $\widehat{\boldsymbol{\beta}}_m$ and $\widehat{\boldsymbol{\beta}}_\infty$ on that example; and the probability of this is exactly $G_{\mathcal{Y}}\left(d\sqrt{\frac{R|\mathcal{Y}|\sigma_0}{\delta m}}\right)$, which upper-bounds $\mathcal{E}(h_m) - \mathcal{E}(h_\infty)$. $\square$

Note that the quantity $\sigma_0$ in Theorem 2 represents the largest value of the inverse (diagonal) Fisher information matrix ([17]). It is thus the smallest value of the (diagonal) Fisher information matrix, and is related to the smallest amount of information one has on the estimation of each parameter $\widehat{\beta}_j^k$. This smallest amount of information is in turn related to the length (in number of occurrences) of the longest (resp. shortest) class in $\mathcal{Y}$ denoted respectively by $n_{max}$ and $n_{min}$ as, the smaller they are, the larger $\sigma_0$ is likely to be.

## 2.3 A learning based node pruning strategy

Let us now consider a hierarchy of classes and a top-down classifier making decisions at each level of the hierarchy. A node-based pruning strategy can be easily derived from the approximation bounds above. Indeed, any node $v$ in the hierarchy $\mathcal{H} = (V, E)$ is associated with three category sets: its sister categories with the node itself $\mathfrak{S}'(v) = \mathfrak{S}(v) \cup \{v\}$, its daughter categories, $\mathfrak{D}(v)$, and the union of its sister and daughter categories, denoted $\mathfrak{F}(v) = \mathfrak{S}(v) \cup \mathfrak{D}(v)$.

These three sets of categories are the ones involved before and after the pruning of node $v$. Let us now denote the MLR classifier by $h_m^{\mathfrak{S}'_v}$ learned from a set of sister categories of node $v$ and the node itself, and by $h_m^{\mathfrak{D}_v}$ a MLR classifier

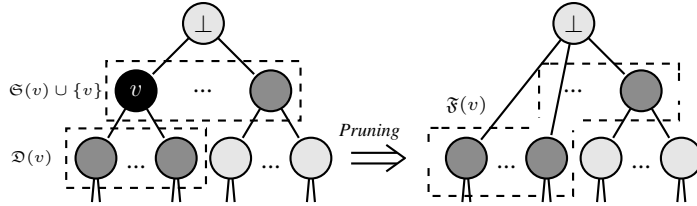

learned from the set of daughter categories of node $v$ ($h_\infty^{\mathfrak{S}'_v}$ and $h_\infty^{\mathfrak{D}_v}$ respectively denote their asymptotic versions). The following theorem is a direct extension of Theorem 2 to this setting.

**Theorem 3** *With the notations defined above, for* MLR *classifiers,* $\forall \epsilon > 0$, $v \in V \setminus \mathcal{Y}$, *one has, with probability at least* $1 - \left( \frac{Rd^2 |\mathfrak{S}'(v)| \sigma_0^{\mathfrak{S}'(v)}}{m_{\mathfrak{S}'(v)} \epsilon^2} + \frac{Rd^2 |\mathfrak{D}(v)| \sigma_0^{\mathfrak{D}(v)}}{m_{\mathfrak{D}(v)} \epsilon^2} \right)$:

$$\mathcal{E}(h_m^{\mathfrak{S}'_v}) + \mathcal{E}(h_m^{\mathfrak{D}_v}) \leq \mathcal{E}(h_\infty^{\mathfrak{S}'_v}) + \mathcal{E}(h_\infty^{\mathfrak{D}_v}) + G_{\mathfrak{S}'(v)}(\epsilon) + G_{\mathfrak{D}(v)}(\epsilon)$$

$\{|\mathcal{Y}^\ell|, m_{\mathcal{Y}^\ell}, \sigma_0^{\mathcal{Y}^\ell}; \mathcal{Y}^\ell \in \{\mathfrak{S}'(v), \mathfrak{D}(v)\}\}$ are constants related to the set of categories $\mathcal{Y}^\ell \in \{\mathfrak{S}'(v), \mathfrak{D}(v)\}$ and involved in the respective bounds stated in Theorem 2. Denoting by $h_m^{\mathfrak{F}_v}$ the MLR classifier trained on the set $\mathfrak{F}(v)$ and by $h_\infty^{\mathfrak{F}_v}$ its asymptotic version, Theorem 3 suggests that one should prune node $v$ if:

$$G_{\mathfrak{F}(v)}(\epsilon) \leq G_{\mathfrak{S}'(v)}(\epsilon) + G_{\mathfrak{D}(v)}(\epsilon) \text{ and } \frac{|\mathfrak{F}(v)| \sigma_0^{\mathfrak{F}(v)}}{m_{\mathfrak{F}(v)}} \leq \frac{|\mathfrak{S}'(v)| \sigma_0^{\mathfrak{S}'(v)}}{m_{\mathfrak{S}'(v)}} + \frac{|\mathfrak{D}(v)| \sigma_0^{\mathfrak{D}(v)}}{m_{\mathfrak{D}(v)}} \quad (5)$$

Furthermore, the bounds obtained rely on the union bound and thus are not likely to be exploitable in practice. They nevertheless exhibit the factors that play an important role in assessing whether a particular trained classifier in the logistic regression family is close or not to its asymptotic version. Each node $v \in V$ can then be characterized by factors in the set $\{|\mathcal{Y}^\ell|, m_{\mathcal{Y}^\ell}, n_{max}^{\mathcal{Y}^\ell}, n_{min}^{\mathcal{Y}^\ell}, G_{\mathcal{Y}^\ell}(.) | \mathcal{Y}^\ell \in \{\mathfrak{S}'(v), \mathfrak{D}(v), \mathfrak{F}(v)\}\}$ which are involved in the estimation of inequalities (5) above. We propose to estimate the confusion term $G_{\mathcal{Y}^\ell}(.)$ with two simple quantities: the average cosine similarity of all the pairs of classes in $\mathcal{Y}^\ell$, and the average symmetric Kullback-Leibler divergences between all the pairs in $\mathcal{Y}^\ell$ of class conditional multinomial distributions.

The procedure for collecting training data associates a positive (resp. negative) class to a node if the pruning of that node leads to a final performance increase (resp. decrease). A meta-classifier is then trained on these features using a training set from a selected class hierarchy. After the learning phase, the meta-classifier is applied to each node of a new hierarchy of classes so as to identify which nodes should be pruned. A simple strategy to adopt is then to prune nodes in sequence: starting from the root node, the algorithm checks which children of a given node $v$ should be pruned by creating the corresponding meta-instance and feeding the meta-classifier; the child that maximizes the probability of the positive class is then pruned; as the set of categories has changed, we recalculate which children of $v$ can be pruned, prune the best one (as above) and iterate this process till no more children of $v$ can be pruned; we then proceed to the children of $v$ and repeat the process.

## 3 Discussion

We start our discussion by presenting results on different hierarchical datasets with different characteristics using MLR and SVM classifiers. The datasets we used in these experiments are two large datasets extracted from the International Patent Classification (**IPC**) dataset[3] and the publicly available DMOZ dataset from the second PASCAL large scale hierarchical text classification challenge (**LSHTC2**)[4]. Both datasets are multi-class; **IPC** is single-label and **LSHTC2** multi-label with an average of 1.02 categories per class. We created 4 datasets from **LSHTC2** by splitting randomly the first layer nodes (11 in total) of the original hierarchy in disjoint subsets. The classes for the **IPC** and **LSHTC2** datasets are organized in a hierarchy in which the documents are assigned to the leaf categories only. Table 1 presents the characteristics of the datasets.

CR denotes the complexity ratio between hierarchical and flat classification, given by the Rademacher complexity term in Theorem 1: $\left( \sum_{v \in V \setminus \mathcal{Y}} |\mathfrak{D}(v)|(|\mathfrak{D}(v)| - 1) \right) / (|\mathcal{Y}|(|\mathcal{Y}| - 1))$; the same constants $B$, $R$ and $L$ are used in the two cases. As one can note, this complexity ratio always goes in favor of the hierarchal strategy, although it is 2 to 10 times higher on the **IPC** dataset, compared to **LSHTC2-1,2,3,4,5**. On the other hand, the ratio of empirical errors (last column of Table 1) obtained with top-down hierarchical classification over flat classification when using SVM

| Dataset | # Tr. | # Test | # Classes | # Feat. | Depth | CR | Error ratio |
|---|---|---|---|---|---|---|---|
| **LSHTC2-1** | 25,310 | 6,441 | 1,789 | 145,859 | 6 | 0.008 | 1.24 |
| **LSHTC2-2** | 50,558 | 13,057 | 4,787 | 271,557 | 6 | 0.003 | 1.32 |
| **LSHTC2-3** | 38,725 | 10,102 | 3,956 | 145,354 | 6 | 0.004 | 2.65 |
| **LSHTC2-4** | 27,924 | 7,026 | 2,544 | 123,953 | 6 | 0.005 | 1.8 |
| **LSHTC2-5** | 68,367 | 17,561 | 7,212 | 192,259 | 6 | 0.002 | 2.12 |
| **IPC** | 46,324 | 28,926 | 451 | 1,123,497 | 4 | 0.02 | 12.27 |

Table 1: Datasets used in our experiments along with the properties: number of training examples, test examples, classes and the size of the feature space, the depth of the hierarchy and the complexity ratio of hierarchical over the flat case ($\sum_{v \in V \setminus \mathcal{Y}} |\mathfrak{D}(v)|(|\mathfrak{D}(v)| - 1)/|\mathcal{Y}|(|\mathcal{Y}| - 1)$), the ratio of empirical error for hierarchical and flat models.

with a linear kernel is this time higher than 1, suggesting the opposite conclusion. The error ratio is furthermore really important on **IPC** compared to **LSHTC2-1,2,3,4,5**. The comparison of the complexity and error ratios on all the datasets thus suggests that the flat classification strategy may be preferred on **IPC**, whereas the hierarchical one is more likely to be efficient on the **LSHTC** datasets. This is indeed the case, as is shown below.

To test our simple node pruning strategy, we learned binary classifiers aiming at deciding whether to prune a node, based on the node features described in the previous section. The label associated to each node in this training set is defined as +1 if pruning the node increases the accuracy of the hierarchical classifier by at least 0.1, and -1 if pruning the node decreases the accuracy by more than 0.1. The threshold at 0.1 is used to avoid too much noise in the training set. The meta-classifier is then trained to learn a mapping from the vector representation of a node (based on the above features) and the labels $\{+1; -1\}$. We used the first two datasets of **LSHTC2** to extract the training data while **LSHTC2-3**, **4**, **5** and **IPC** were employed for testing.

The procedure for collecting training data is repeated for the MLR and SVM classifiers resulting in three meta-datasets of 119 (19 positive and 100 negative), 89 (34 positive and 55 negative) and 94 (32 positive and 62 negative) examples respectively. For the binary classifiers, we used AdaBoost with random forest as a base classifier, setting the number of trees to 20, 50 and 50 for the MLR and SVM classifiers respectively and leaving the other parameters at their default values. Several values have been tested for the number of trees ($\{10, 20, 50, 100 \text{ and } 200\}$), the depth of the trees ($\{\text{unrestricted}, 5, 10, 15, 30, 60\}$), as well as the number of iterations in AdaBoost ($\{10, 20, 30\}$). The final values were selected by cross-validation on the training set (**LSHTC2-1** and **LSHTC2-2**) as the ones that maximized accuracy and minimized false-positive rate in order to prevent degradation of accuracy.

We compare the fully flat classifier (FL) with the fully hierarchical (FH) top-down *Pachinko* machine, a random pruning (RN) and the proposed pruning method (PR). For the random pruning we restrict the procedure to the first two levels and perform 4 random prunings (this is the average number of prunings that are performed in our approach). For each dataset we perform 5 independent runs for the random pruning and we record the best performance. For MLR and SVM, we use the LibLinear library [8] and apply the $L2$-regularized versions, setting the penalty parameter $C$ by cross-validation.

The results on **LSHTC2-3,4,5** and **IPC** are reported in Table 2. On all **LSHTC** datasets flat classification performs worse than the fully hierarchy top-down classification, for all classifiers. These results are in line with complexity and empirical error ratios for SVM estimated on different collections and shown in table 1 as well as with the results obtained in [14, 7] over the same type of taxonomies. Further, the work by [14] demonstrated that class hierarchies on **LSHTC** datasets suffer from *rare categories* problem, i.e., 80% of the target categories in such hierarchies have less than 5 documents assigned to them.

As a result, flat methods on such datasets face unbalanced classification problems which results in smaller error ratios; hierarchical classification should be preferred in this case. On the other hand, for hierarchies such as the one of **IPC**, which are relatively well balanced and do not suffer from the rare categories phenomenon, flat classification performs at par or even better than hierarchical

| | LSHTC2-3 | | LSHTC2-4 | | LSHTC2-5 | | IPC | |
|---|---|---|---|---|---|---|---|---|
| | MLR | SVM | MLR | SVM | MLR | SVM | MLR | SVM |
| FL | $0.528^{\downarrow\downarrow}$ | $0.535^{\downarrow\downarrow}$ | $0.497^{\downarrow\downarrow}$ | $0.501^{\downarrow\downarrow}$ | $0.542^{\downarrow\downarrow}$ | $0.547^{\downarrow\downarrow}$ | 0.546 | **0.446** |
| RN | $0.493^{\downarrow\downarrow}$ | $0.517^{\downarrow\downarrow}$ | $0.478^{\downarrow\downarrow}$ | $0.484^{\downarrow\downarrow}$ | $0.532^{\downarrow\downarrow}$ | $0.536^{\downarrow}$ | $0.547^{\downarrow}$ | $0.458^{\downarrow\downarrow}$ |
| FH | $0.484^{\downarrow\downarrow}$ | $0.498^{\downarrow\downarrow}$ | $0.473^{\downarrow\downarrow}$ | $0.476^{\downarrow}$ | $0.526^{\downarrow}$ | 0.527 | $0.552^{\downarrow}$ | $0.465^{\downarrow\downarrow}$ |
| PR | **0.480** | **0.493** | **0.469** | **0.472** | **0.522** | **0.523** | **0.544** | 0.450 |

Table 2: Error results across all datasets. Bold typeface is used for the best results. Statistical significance (using micro sign test (s-test) as proposed in [20]) is denoted with $\downarrow$ for p-value$<0.05$ and with $\downarrow\downarrow$ for p-value$<0.01$.

classification. This is in agreement with the conclusions obtained in recent studies, as [2, 9, 16, 6], in which the datasets considered do not have *rare categories* and are more well-balanced.

The proposed hierarchy pruning strategy aims to adapt the given taxonomy structure for better classification while maintaining the ancestor-descendant relationship between a given pair of nodes. As shown in Table 2, this simple learning based pruning strategy leads to statistically significant better results for all three classifiers compared to both the original taxonomy and a randomly pruned one. A similar result is reported in [18] through a pruning of an entire layer of the hierarchy, which can be seen as a generalization, even though empirical in nature, of the pruning strategy retained here. Another interesting approach to modify the original taxonomy is presented in [21]. In this study, three other elementary modification operations are considered, again with an increase of performance.

## 4 Conclusion

We have studied in this paper flat and hierarchical classification strategies in the context of large-scale taxonomies, through error generalization bounds of multiclass, hierarchical classifiers. The first theorem we have introduced provides an explanation to several empirical results related to the performance of such classifiers. We have also introduced a well-founded way to simplify a taxonomy by selectively pruning some of its nodes, through a meta-classifier. The features retained in this meta-classifier derive from the error generalization bounds we have proposed. The experimental results reported here (as well as in other papers) are in line with our theoretical developments and justify the pruning strategy adopted.

This is the first time, to our knowledge, that a data dependent error generalization bound is proposed for multiclass, hierarchical classifiers and that a theoretical explanation is provided for the performance of flat and hierarchical classification strategies in large-scale taxonomies. In particular, there is, up to now, no consensus on which classification scheme, flat or hierarchical, to use on a particular category system. One of our main conclusions is that top-down hierarchical classifiers are well suited to unbalanced, large-scale taxonomies, whereas flat ones should be preferred for well-balanced taxonomies.

Lastly, our theoretical development also suggests possibilities to grow a hierarchy of classes from a (large) set of categories, as has been done in several studies (*e.g.* [2]). We plan to explore this in future work.

## 5 Acknowledgments

This work was supported in part by the ANR project Class-Y, the Mastodons project Garguantua, the LabEx PERSYVAL-Lab ANR-11-LABX-0025 and the European project BioASQ (grant agreement no. 318652).

## Footnotes

[1] `www.dmoz.org`

[2] The study in [19] introduces a slightly different simplification, through an embedding of both categories and documents into a common space.

[3]http://www.wipo.int/classifications/ipc/en/support/

[4]http://lshtc.iit.demokritos.gr/

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
