[Supplementary Material]

# On Flat versus Hierarchical Classification in Large-Scale Taxonomies

**Rohit Babbar, Ioannis Partalas, Eric Gaussier, Massih-Reza Amini**
Université Joseph Fourier, Laboratoire Informatique de Grenoble
BP 53 - F-38041 Grenoble Cedex 9
`firstname.lastname@imag.fr`

## Appendix: Proof of Theorem 1

Let us first recall Theorem 1:

**Theorem 1** *Let $\mathcal{S} = ((\mathbf{x}^{(i)}, y^{(i)}))_{i=1}^m$ be a dataset of $m$ examples drawn i.i.d. according to a probability distribution $\mathcal{D}$ over $\mathcal{X} \times \mathcal{Y}$, and let $\mathcal{A}$ be a Lipschitz function with constant $L$ dominating the $0/1$ loss; further let $K : \mathcal{X} \times \mathcal{X} \to \mathbb{R}$ be a PDS kernel and let $\Phi : \mathcal{X} \to \mathbb{H}$ be the associated feature mapping function. Assume that there exists $R > 0$ such that $K(\mathbf{x}, \mathbf{x}) \le R^2$ for all $\mathbf{x} \in \mathcal{X}$. Then, for all $1 < \delta < 0$, with probability at least $(1 - \delta)$ the following hierarchical multiclass classification generalization bound holds for all $g_f \in \mathcal{G}_{\mathcal{F}_B}$ :*

$$\mathcal{E}(g_f) \le \frac{1}{m} \sum_{i=1}^m \mathcal{A}(g_f(\mathbf{x}^{(i)}, y^{(i)})) + \frac{8BRL}{\sqrt{m}} \sum_{v \in V \setminus \mathcal{Y}} |\mathfrak{D}(v)|(|\mathfrak{D}(v)| - 1) + 3\sqrt{\frac{\ln(2/\delta)}{2m}} \quad (1)$$

*where, $\mathcal{G}_{\mathcal{F}_B} = \{(\mathbf{x}, y) \in \mathcal{X} \times \mathcal{Y} \mapsto \min_{v \in \mathfrak{P}(y)}(f(\mathbf{x}, v) - \max_{v' \in \mathfrak{S}(v)} f(\mathbf{x}, v')) \mid f \in \mathcal{F}_B\}$, $\mathcal{F}_B = \{(\mathbf{x}, v) \in \mathcal{X} \times V \mapsto \langle \mathbf{w}_v, x \rangle \mid \mathbf{W} = (w_1 \dots, w_{|V|}), \|\mathbf{W}\|_{\mathbb{H}} \le B\}$, and $|\mathfrak{D}(v)|$ denotes the number of daughters of node $v$.*

**Proof**     Exploiting the fact that $\mathcal{A}$ dominates the $0/1$ loss and using the Rademacher data-dependent generalization bound presented in Theorem 4.9 of [2], one has:

$$
\begin{aligned}
\mathbb{E}_{(x,y) \sim \mathcal{D}} \left[ \mathbf{1}_{g_f(\mathbf{x},y) \le 0} - 1 \right] &\le \mathbb{E}_{(\mathbf{x},y) \sim \mathcal{D}} \left[ \mathcal{A} \circ g_f(\mathbf{x}, y) - 1 \right] \\
&\le \frac{1}{m} \sum_{i=1}^m (\mathcal{A}(g_f(\mathbf{x}^{(i)}, y^{(i)})) - 1) + \hat{\mathcal{R}}_m((\mathcal{A} - 1) \circ \mathcal{G}_{\mathcal{F}_B}, \mathcal{S}) + 3\sqrt{\frac{\ln(2/\delta)}{2m}}
\end{aligned}
$$

where $\hat{\mathcal{R}}_m$ denotes the empirical Rademacher complexity of $(\mathcal{A} - 1) \circ \mathcal{G}_{\mathcal{F}_B}$ on $\mathcal{S}$. As $x \mapsto \mathcal{A}(x)$ is a Lipschtiz function with constant $L$ and $(\mathcal{A} - 1)(0) = 0$, we further have:

$$\hat{\mathcal{R}}_m((\mathcal{A} - 1) \circ \mathcal{G}_{\mathcal{F}_B}, \mathcal{S}) \le 2L \hat{\mathcal{R}}_m(\mathcal{G}_{\mathcal{F}_B}, \mathcal{S})$$

with:

$$
\begin{aligned}
\hat{\mathcal{R}}_m(\mathcal{G}_{\mathcal{F}_B}, \mathcal{S}) &= \mathbb{E}_\sigma \left[ \sup_{g_f \in \mathcal{G}_{\mathcal{F}_B}} \left| \frac{2}{m} \sum_{i=1}^m \sigma_i \, g_f(\mathbf{x}^{(i)}, y^{(i)}) \right| \right] \\
&= \mathbb{E}_\sigma \left[ \sup_{f \in \mathcal{F}_B} \left| \frac{2}{m} \sum_{i=1}^m \sigma_i \min_{v \in \mathfrak{P}(y^{(i)})} (f(\mathbf{x}^{(i)}, v) - \max_{v' \in \mathfrak{S}(v)} f(\mathbf{x}^{(i)}, v')) \right| \right]
\end{aligned}
$$

Let us define the mapping $c$ from $\mathcal{F}_B \times \mathcal{X} \times \mathcal{Y}$ into $V \times V$ as:

$$
\begin{aligned}
c(f, \mathbf{x}, y) = (v, v') &\Rightarrow (f(\mathbf{x}, v') = \max_{v'' \in \mathfrak{S}(v)} f(\mathbf{x}, v'')) \\
&\wedge (f(\mathbf{x}, v) - f(\mathbf{x}, v') = \min_{u \in \mathfrak{P}(y)} (f(\mathbf{x}, u) - \max_{u' \in \mathfrak{S}(u)} f(\mathbf{x}, u')))
\end{aligned}
$$

This definition is similar to the one given in [1] for flat multiclass classification. Then, by construction of $c$:

$$\hat{\mathcal{R}}_m(\mathcal{G}_{\mathcal{F}_B}, \mathcal{S}) \leq \frac{2}{m} \mathbb{E}_\sigma \left[ \sup_{f \in \mathcal{F}_B} \sum_{(v,v') \in V^2, v' \in \mathfrak{S}(v)} \left| \sum_{i:c(f,\mathbf{x}^{(i)},y^{(i)})=(v,v')} \sigma_i(f(\mathbf{x}^{(i)},v) - f(\mathbf{x}^{(i)},v')) \right| \right]$$

By definition, $f(\mathbf{x}^{(i)}, v) - f(\mathbf{x}^{(i)}, v') = \langle \mathbf{w}_v - \mathbf{w}_{v'}, \Phi(\mathbf{x}^{(i)}) \rangle$ and using Cauchy-Schwartz inequality:

$$
\begin{aligned}
\hat{\mathcal{R}}_m(\mathcal{G}_{\mathcal{F}_B}, \mathcal{S}) &\leq \frac{2}{m} \mathbb{E}_\sigma \left[ \sup_{\|\mathbf{W}\|_{\mathbb{H}} \leq B} \sum_{(v,v') \in V^2, v' \in \mathfrak{S}(v)} \left| \left\langle \mathbf{w}_v - \mathbf{w}_{v'}, \sum_{i:c(f,\mathbf{x}^{(i)},y^{(i)})=(v,v')} \sigma_i \Phi(\mathbf{x}^{(i)}) \right\rangle \right| \right] \\
&\leq \frac{2}{m} \mathbb{E}_\sigma \left[ \sup_{\|\mathbf{W}\|_{\mathbb{H}} \leq B} \sum_{(v,v') \in V^2, v' \in \mathfrak{S}(v)} \|\mathbf{w}_v - \mathbf{w}_{v'}\|_{\mathbb{H}} \left\| \sum_{i:c(f,\mathbf{x}^{(i)},y^{(i)})=(v,v')} \sigma_i \Phi(\mathbf{x}^{(i)}) \right\|_{\mathbb{H}} \right] \\
&\leq \frac{4B}{m} \sum_{(v,v') \in V^2, v' \in \mathfrak{S}(v)} \mathbb{E}_\sigma \left[ \left\| \sum_{i:c(f,\mathbf{x}^{(i)},y^{(i)})=(v,v')} \sigma_i \Phi(\mathbf{x}^{(i)}) \right\|_{\mathbb{H}} \right]
\end{aligned}
$$

Using Jensen's inequality, and as, $\forall i,j \in \{l|c(f,\mathbf{x}^{(l)},y^{(l)}) = (v,v')\}^2, i \neq j, \mathbb{E}_\sigma[\sigma_i \sigma_j] = 0$, we get:

$$
\begin{aligned}
\hat{\mathcal{R}}_m(\mathcal{G}_{\mathcal{F}_B}, \mathcal{S}) &\leq \frac{4B}{m} \sum_{(v,v') \in V^2, v' \in \mathfrak{S}(v)} \left( \mathbb{E}_\sigma \left[ \left\| \sum_{i:c(f,\mathbf{x}^{(i)},y^{(i)})=(v,v')} \sigma_i \Phi(\mathbf{x}^{(i)}) \right\|_{\mathbb{H}}^2 \right] \right)^{1/2} \\
&= \frac{4B}{m} \sum_{(v,v') \in V^2, v' \in \mathfrak{S}(v)} \left( \sum_{i:c(f,\mathbf{x}^{(i)},y^{(i)})=(v,v')} \left\| \Phi(\mathbf{x}^{(i)}) \right\|_{\mathbb{H}}^2 \right)^{1/2} \\
&= \frac{4B}{m} \sum_{(v,v') \in V^2, v' \in \mathfrak{S}(v)} \left( \sum_{i:c(f,\mathbf{x}^{(i)},y^{(i)})=(v,v')} K\left(\mathbf{x}^{(i)}, \mathbf{x}^{(i)}\right) \right)^{1/2} \\
&\leq \frac{4B}{m} \sum_{(v,v') \in V^2, v' \in \mathfrak{S}(v)} \left( mR^2 \right)^{1/2} \\
&= \frac{4BR}{\sqrt{m}} \sum_{v \in V \setminus \mathcal{Y}} |\mathfrak{D}(v)|(|\mathfrak{D}(v)| - 1)
\end{aligned}
$$

Plugging this bound into the first inequality yields the desired result. $\square$