[Reviews · NeurIPS 2013]

Submitted by Assigned_Reviewer_5

Summary: The paper concerns hierarchical multi-class classification. The authors study whether and when a hierarchical classifier can be more beneficial than its flat counterpart. They proof a generalization bound that provides an explanation when a flat and when a hierarchical classifier should be used. Additionally, the authors provide an approach for logistic regression and naive Bayes classifiers, which enables pruning of nodes in large-scale hierarchies.

Quality: The authors consider a very interesting and up-to-date problem. Therefore I was very glad to read this paper. The first bound obtained by the authors is very interesting and indeed provides an explanation of existing empirical results. The rest of the paper is less clear. It seems that Lemma 1 is quite similar to standard results in statistics. At least, I suppose that a similar result has been published elsewhere. I am also not entirely convinced by the pruning strategy introduced by the authors. A better explanation is required here. It is also not clear why AdaBoost with Random Forest is used as the meta-classifier. What are the reasons to use such a complex classifier here, while the main classification problem is solved by rather simple linear algorithms? It is also not clear how the results concerning pruning of the hierarchy can be generalized to other algorithms.

Clarity: The paper is quite dense, therefore not easy to read. It contains many theoretical results, which are not easy to verify in the limited time of the reviewing period. Personally, I would prefer a paper that would be constructed around one of the results, for example, the first theorem. Then the verification of the results would be easier, as well as the reading of the paper. Unfortunately, the second part of the paper concerning logistic regression and pruning is not so clear as the first part. The description of the meta-classifier could be improved. It seems that there is missing something around Theorem 3. At least it is not clear when Theorem 3 ends.

Originality: The results are in my opinion very original. I do not know any paper that considers a similar problem.

Significance: The obtained results can be very significant. At least the first theorem (if the proof is correct) gives a nice explanation of the performance of flat and hierarchical classifiers.
Summary: Summary: It is an interesting paper that concerns an up-to-date problem. Personally, I like very much the first part of the paper around theorem 1. The rest of the paper is unfortunately less understandable.

Update (after rebuttal): I thank the authors for their clarifications. The second part of the paper is now a little bit clearer for me, however, still I think that this part could be written in a more comprehensible way.

Submitted by Assigned_Reviewer_6

This paper investigates a tradeoff in hierarchical classification: when a hierarchy is deep, a classifier needs to make a decision at every level of the tree which is likely to propagate errors. When a hierarchy is flat(ter), less decision need to be made so there is less room for errors to propagate, however, the decisions are harder (because the cardinality of is higher).

This paper introduces a data-dependent generalization error bound for kernel based hypotheses. The main theorem of the paper states an upper bound on the generalization error of a hierarchical classifier in terms of the empirical error and the Rademacher complexity of the classifier. The former encouraging flat classifiers, the latter encouraging deep classifiers.

The paper uses the insight from the generalization bound to come up with a strategy to prune hierarchical classifiers. The paragraph starting on line 307 was a bit unclear to me; I missed the motivation of the methodology using the metaclassifier and how it relates to improvements in the generalization bound.

The paper is clearly written and solves an interesting and important problem. On the theoretical side, the paper contributes to the literature on hierarchical classification. My main issue with the paper is that it is unclear how to use their insights for anyone doing hierarchical classification. I think the authors can do a better job of describing the practical take-away around how to best do pruning of a hierarchy to improve hierarchical classification.
Summary: A well written theoretical paper on the important topic of hierarchical classification. The writeup falls short on taking the theoretical insight and deriving practical procedures for improving hierarchical classification.

Submitted by Assigned_Reviewer_7

Summary
=======

This paper proposes a bound on the generalization error of hierarchical multi-classifiers for large-scale taxonomies. Such bound provides a justification of empirical results of flat and hierarchical classifiers. The paper also proposes a bound on the approximation error of a family of classifiers. This is used to define a node-classifier for pruning large-scale taxonomy and thus improving the overall accuracy. Some experiments illustrate the findings of the proposed theory on two well-known taxonomies.

Evaluation
========

The paper is well written and presented.
It contains interesting technical content, which appears to be sound.
In general the paper is of high quality.

However, this reviewer sees two limitations of the proposed methods:

1. Considering hierarchical classification as the task of only classifying the leaf nodes is rather limitative for two reasons:

a. In real cases, category nodes can contain documents that do not belong to any of their children. This is very common as, when new documents of new types have to be classified, there is no specific child node to accommodate them. Thus the best choice is to assign them to the father nodes.

b. This reviewer is not so sure that there is an actual debate on using flat and hierarchical models in the setting proposed by the authors (i.e., only leaf classification). Indeed, when working in the authors' setting, the flat model seems superior.

In contrast, top-down methods may get better accuracy when classification is also carried out in the internal nodes. For example, the following paper clearly shows that the top-down method is more accurate than the flat one in such setting:

Alessandro Moschitti, Qi Ju, Richard Johansson: Modeling Topic Dependencies in Hierarchical Text Categorization. ACL 2012: 759-767.

In this respect a close comparison with other methods, e.g., the one above, those cited by the authors (e.g., Gopal et al.) and the following

Zhou, D., Xiao, L., Wu, M.: Hierarchical classification via orthogonal transfer. ICML11.

would be appreciated.

2. The proposed bounds do not look very strict: the Rademacher complexity term is a rough approximation and does not consider feature distribution/relevance, which has a major impact in text categorization.
For example, in Reuters 21578, there are very rare categories. They may contain about 0.1% of the entire training data, i.e., about 10 documents but, for them, systems can reach accuracy of about 90%, larger than for other more populated categories.
This suggests that the data imbalance of categories is an important factor but it is not enough. For example, also category ambiguity (how much a category is similar to others) should be considered.


After the authors' response
====================

In addition to the reviewer comments, the concepts expressed in the authors' answer should be included in the final version of the paper.
Moreover, the authors' answer does not completely solve all the reviewer's doubts, thus the authors should inform the reader about the possible theory's flaws.
Summary: - The technical content is very good but the proposed bounds may result not enough strict. The authors should inform the reader about this.
- The paper claims should be lowered in strength and remodulated according to the reviewers' comments and the authors' response.
Author Feedback

Author rebuttal: We would like to thank the reviewers for making several comments and suggestions.

# Replies to common concerns of reviewers 1 and 2

The first part of our paper aims at providing a general explanation on the behavior of flat and hierarchical approaches. This general explanation, based on the Rademacher complexity, does not directly provide a practical procedure to improve classification in large-scale taxonomies. The second part of our paper aims at doing so by supplying an additional analysis of the approximation error of two well-known classifiers: logistic regression and Naive Bayes. From this additional analysis, we derive features, compliant with the Rademacher analysis, from which a practical procedure to improve a given taxonomy is proposed. As illustrated in the experiments, this procedure leads to a new taxonomy with improved classification accuracy.

In sum, the overall approach we have designed allows one to modify a given taxonomy so as to improve the accuracy of top-down classifiers deployed on it; we agree that the second part of the paper is a bit more complex, but it allows to bridge between the theoretical analysis shown before and a practical simple pruning strategy.

We have tried several non-linear classifiers performing well with the few meta-few features we deduced from the approximation bound, and boosting with random forest was the best performing model. We will provide additional results we obtained using a multi-layer perceptron with one hidden layer and SVM with non-linear kernels in the supplementary material, in the case where the paper is accepted.

Lemma 1 is indeed an extension, to the multiclass case, of a known result (our ref. 6) and definitely resembles the result stated there and elsewhere; this extension is new as far as we can tell.

# Replies to concerns of reviewer 3

1(a) The mandatory leaf node classification setting described at the beginning of section 2 is a general case which accommodates the scenario in which there are documents at an internal node that do not belong to any of its child nodes. In order to tackle this problem, an extra leaf node is assumed at that internal node and all such documents are assigned to this extra leaf node. There is no loss of generality using this transformation of the hierarchy and the classification problem remains well-posed. This approach has been used in many other works on hierarchical classification, as "Distribution-calibrated hierarchical classification", O. Dekel., NIPS 2010. Another recent work which highlights this setting is "Mandatory Leaf Node Prediction in Hierarchical Multilabel Classification", Bi et al. NIPS 2012.

1(b) Please note that Table 2 exhibits error rates (and not accuracy values) of various classification schemes. The results reported in this table show that the hierarchical top-down approach is indeed better for all datasets, except the IPC dataset wherein there are no rare categories.

1(c) There is in fact a strong correlation between various (flat and/or hierarchical) evaluation measures. The results we obtained do not change, for example, if we use tree induced error or the F1-score instead of the accuracy. We will add these two measures in the final version of the paper, provided it is accepted.

1(d) The hierarchical classification technique has computational advantages in terms of training and prediction time over flat classification. For classification accuracy, our data-dependent generalization error analysis can help in identifying scenarios in which one technique is preferred over the other. This has important implications in designing practical large scale classification techniques under the constraints of higher accuracy, and faster training and prediction speed.

2(a) Theorem 1 and the explanation given thereafter in section 2.1 provide a general explanation for the classification accuracy behavior observed in hierarchical classification. Although this bound does not explicitly exhibit all magnitudes implicated in multi-class hierarchical classification, it points out simple cases where the hierarchical classification may be preferred to the flat classification scenario and vice versa. The asymptotic bound presented in section 2.2 points out more specific features such as the degree of inter-class overlap. In the experimental part, we characterized this specific confusion feature by the KL divergence and the cosine similarity measures.

2(b) The LSHTC datasets used in the experiments differ from the Reuters dataset in many respects ; specially the number of target classes that are of the order of thousands in this case compared to Reuters RCV1/RCV2 collections (of the order of one hundred). Furthermore, as compared to the Reuters dataset, the rarity phenomenon of target classes in LSHTC datasets is much more extreme since many such classes may just have 2 to 4 training documents making it quite hard to learn an accurate classifier identifying these classes in a flat classification scenario.